# Efficient Learning of Continuous-Time Hidden Markov Models for Disease Progression

**Yu-Ying Liu, Shuang Li, Fuxin Li, Le Song, and James M. Rehg**
College of Computing
Georgia Institute of Technology
Atlanta, GA

## Abstract

The Continuous-Time Hidden Markov Model (CT-HMM) is an attractive approach to modeling disease progression due to its ability to describe noisy observations arriving irregularly in time. However, the lack of an efficient parameter learning algorithm for CT-HMM restricts its use to very small models or requires unrealistic constraints on the state transitions. In this paper, we present the first complete characterization of efficient EM-based learning methods for CT-HMM models. We demonstrate that the learning problem consists of two challenges: the estimation of posterior state probabilities and the computation of end-state conditioned statistics. We solve the first challenge by reformulating the estimation problem in terms of an equivalent discrete time-inhomogeneous hidden Markov model. The second challenge is addressed by adapting three approaches from the continuous time Markov chain literature to the CT-HMM domain. We demonstrate the use of CT-HMMs with more than 100 states to visualize and predict disease progression using a glaucoma dataset and an Alzheimer's disease dataset.

## 1 Introduction

The goal of disease progression modeling is to learn a model for the temporal evolution of a disease from sequences of clinical measurements obtained from a longitudinal sample of patients. By distilling population data into a compact representation, disease progression models can yield insights into the disease process through the visualization and analysis of disease trajectories. In addition, the models can be used to predict the future course of disease in an individual, supporting the development of individualized treatment schedules and improved treatment efficiencies. Furthermore, progression models can support phenotyping by providing a natural similarity measure between trajectories which can be used to group patients based on their progression.

Hidden variable models are particularly attractive for modeling disease progression for three reasons: 1) they support the abstraction of a disease state via the latent variables; 2) they can deal with noisy measurements effectively; and 3) they can easily incorporate dynamical priors and constraints. While conventional hidden Markov models (HMMs) have been used to model disease progression, they are not suitable in general because they assume that measurement data is sampled regularly at discrete intervals. However, in reality patient visits are *irregular* in time, as a consequence of scheduling issues, missed visits, and changes in symptomatology.

A *Continuous-Time* HMM (CT-HMM) is an HMM in which both the transitions between hidden states and the arrival of observations can occur at arbitrary (continuous) times [1, 2]. It is therefore suitable for irregularly-sampled temporal data such as clinical measurements [3, 4, 5]. Unfortunately, the additional modeling flexibility provided by CT-HMM comes at the cost of a more complex inference procedure. In CT-HMM, not only are the hidden states unobserved, but the *transition times* at which the hidden states are changing are also unobserved. Moreover, multiple unobserved hidden state transitions can occur between two successive observations. A previous method addressed these challenges by directly maximizing the data likelihood [2], but this approach is limited

to very small model sizes. A general EM framework for continuous-time dynamic Bayesian networks, of which CT-HMM is a special case, was introduced in [6], but that work did not address the question of efficient learning. Consequently, there is a need for efficient CT-HMM learning methods that can scale to large state spaces (e.g. hundreds of states or more) [7].

A key aspect of our approach is to leverage the existing literature for continuous time Markov chain (CTMC) models [8, 9, 10]. These models assume that states are directly observable, but retain the irregular distribution of state transition times. EM approaches to CTMC learning compute the expected state durations and transition counts conditioned on each pair of successive observations. The key computation is the evaluation of integrals of the matrix exponential (Eqs. 12 and 13). Prior work by Wang et. al. [5] used a closed form estimator due to [8] which assumes that the transition rate matrix can be diagonalized through an eigendecomposition. Unfortunately, this is frequently not achievable in practice, limiting the usefulness of the approach. We explore two additional CTMC approaches [9] which use (1) an alternative matrix exponential on an auxillary matrix (*Expm* method); and (2) a direct truncation of the infinite sum expansion of the exponential (*Unif* method). Neither of these approaches have been previously exploited for CT-HMM learning.

We present the first comprehensive framework for efficient EM-based parameter learning in CT-HMM, which both extends and unifies prior work on CTMC models. We show that a CT-HMM can be conceptualized as a time-inhomogenous HMM which yields posterior state distributions at the observation times, coupled with CTMCs that govern the distribution of hidden state transitions between observations (Eqs. 9 and 10). We explore both soft (forward-backward) and hard (Viterbi decoding) approaches to estimating the posterior state distributions, in combination with three methods for calculating the conditional expectations. We validate these methods in simulation and evaluate our approach on two real-world datasets for glaucoma and Alzheimer's disease, including visualizations of the progression model and predictions of future progression. Our approach outperforms a state-of-the-art method [11] for glaucoma prediction, which demonstrates the practical utility of CT-HMM for clinical data modeling.

## 2 Continuous-Time Markov Chain

A continuous-time Markov chain (CTMC) is defined by a finite and discrete state space $S$, a state transition rate matrix $Q$, and an initial state probability distribution $\pi$. The elements $q_{ij}$ in $Q$ describe the rate the process transitions from state $i$ to $j$ for $i \neq j$, and $q_{ii}$ are specified such that each row of $Q$ sums to zero ($q_i = \sum_{j \neq i} q_{ij}$, $q_{ii} = -q_i$) [1]. In a time-homogeneous process, in which the $q_{ij}$ are independent of $t$, the sojourn time in each state $i$ is exponentially-distributed with parameter $q_i$, which is $f(t) = q_i e^{-q_i t}$ with mean $1/q_i$. The probability that the process's next move from state $i$ is to state $j$ is $q_{ij}/q_i$. When a realization of the CTMC is *fully* observed, meaning that one can observe every transition time $(t'_0, t'_1, \ldots, t'_{V'})$, and the corresponding state $Y' = \{y_0 = s(t'_0), \ldots, y_{V'} = s(t'_{V'})\}$, where $s(t)$ denotes the state at time $t$, the complete likelihood (CL) of the data is

$$CL = \prod_{v'=0}^{V'-1} (q_{y_{v'},y_{v'+1}}/q_{y_{v'}})(q_{y_{v'}} e^{-q_{y_{v'}} \tau_{v'}}) = \prod_{v'=0}^{V'-1} q_{y_{v'},y_{v'+1}} e^{-q_{y_{v'}} \tau_{v'}} = \prod_{i=1}^{|S|} \prod_{j=1,j\neq i}^{|S|} q_{ij}^{n_{ij}} e^{-q_i \tau_i} \quad (1)$$

where $\tau_{v'} = t'_{v'+1} - t'_{v'}$ is the time interval between two transitions, $n_{ij}$ is the number of transitions from state $i$ to $j$, and $\tau_i$ is the total amount of time the chain remains in state $i$.

In general, a realization of the CTMC is observed only at *discrete and irregular* time points $(t_0, t_1, \ldots, t_V)$, corresponding to a state sequence $Y$, which are *distinct* from the switching times. As a result, the Markov process between two consecutive observations is *hidden*, with potentially many unobserved state transitions. Thus both $n_{ij}$ and $\tau_i$ are unobserved. In order to express the likelihood of the incomplete observations, we can utilize a discrete time hidden Markov model by defining a state transition probability matrix for each distinct time interval $t$, $P(t) = e^{Qt}$, where $P_{ij}(t)$, the entry $(i,j)$ in $P(t)$, is the probability that the process is in state $j$ after time $t$ given that it is in state $i$ at time 0. This quantity takes into account all possible intermediate state transitions and timing between $i$ and $j$ which are not observed. Then the likelihood of the data is

$$L = \prod_{v=0}^{V-1} P_{y_v,y_{v+1}}(\tau_v) = \prod_{v=0}^{V-1} \prod_{i,j=1}^{|S|} P_{ij}(\tau_v)^{\mathbb{I}(y_v=i,y_{v+1}=j)} = \prod_{\Delta=1}^{r} \prod_{i,j=1}^{|S|} P_{ij}(\tau_\Delta)^{C(\tau=\tau_\Delta,y_v=i,y_{v+1}=j)} \quad (2)$$

where $\tau_v = t_{v+1} - t_v$ is the time interval between two observations, $\mathbb{I}(y_v = i, y_{v+1} = j)$ is an indicator function that is 1 if the condition is true, otherwise it is 0, $\tau_\Delta$, $\Delta = 1, \ldots, r$, represents $r$ unique values among all time intervals $\tau_v$, and $C(\tau = \tau_\Delta, y_v = i, y_{v+1} = j)$ is the total counts

from all successive visits when the condition is true. Note that there is no analytic maximizer of $L$, due to the structure of the matrix exponential, and direct numerical maximization with respect to $Q$ is computationally challenging. This motivates the use of an EM-based approach.

An EM algorithm for CTMC is described in [8]. Based on Eq. 1, the expected complete log likelihood takes the form $\sum_{i=1}^{|S|} \sum_{j=1, j \neq i}^{|S|} \{\log(q_{ij}) \mathbb{E}[n_{ij}|Y, \hat{Q}_0] - q_i \mathbb{E}[\tau_i|Y, \hat{Q}_0]\}$, where $\hat{Q}_0$ is the current estimate for $Q$, and $\mathbb{E}[n_{ij}|Y, \hat{Q}_0]$ and $\mathbb{E}[\tau_i|Y, \hat{Q}_0]$ are the expected state transition count and total duration given the incomplete observation $Y$ and the current transition rate matrix $\hat{Q}_0$, respectively. Once these two expectations are computed in the E-step, the updated $\hat{Q}$ parameters can be obtained via the M-step as

$$\hat{q}_{ij} = \frac{\mathbb{E}[n_{ij}|Y, \hat{Q}_0]}{\mathbb{E}[\tau_i|Y, \hat{Q}_0]}, i \neq j \quad \text{and} \quad \hat{q}_{ii} = -\sum_{j \neq i} \hat{q}_{ij}. \tag{3}$$

Now the main computational challenge is to evaluate $\mathbb{E}[n_{ij}|Y, \hat{Q}_0]$ and $\mathbb{E}[\tau_i|Y, \hat{Q}_0]$. By exploiting the properties of the Markov process, the two expectations can be decomposed as [12]:

$$\mathbb{E}[n_{ij}|Y, \hat{Q}_0] = \sum_{v=0}^{V-1} \mathbb{E}[n_{ij}|y_v, y_{v+1}, \hat{Q}_0] = \sum_{v=0}^{V-1} \sum_{k,l=1}^{|S|} \mathbb{I}(y_v = k, y_{v+1} = l) \mathbb{E}[n_{ij}|y_v = k, y_{v+1} = l, \hat{Q}_0]$$

$$\mathbb{E}[\tau_i|Y, \hat{Q}_0] = \sum_{v=0}^{V-1} \mathbb{E}[\tau_i|y_v, y_{v+1}, \hat{Q}_0] = \sum_{v=0}^{V-1} \sum_{k,l=1}^{|S|} \mathbb{I}(y_v = k, y_{v+1} = l) \mathbb{E}[\tau_i|y_v = k, y_{v+1} = l, \hat{Q}_0]$$

where $\mathbb{I}(y_v = k, y_{v+1} = l) = 1$ if the condition is true, otherwise it is 0. Thus, the computation reduces to computing the end-state conditioned expectations $\mathbb{E}[n_{ij}|y_v = k, y_{v+1} = l, \hat{Q}_0]$ and $\mathbb{E}[\tau_i|y_v = k, y_{v+1} = l, \hat{Q}_0]$, for all $k, l, i, j \in S$. These expectations are also a key step in CT-HMM learning, and Section 4 presents our approach to computing them.

## 3 Continuous-Time Hidden Markov Model

In this section, we describe the continuous-time hidden Markov model (CT-HMM) for disease progression and the proposed framework for CT-HMM learning.

### 3.1 Model Description

In contrast to CTMC, where the states are directly observed, none of the states are directly observed in CT-HMM. Instead, the available observational data $o$ depends on the hidden states $s$ via the measurement model $p(o|s)$. In contrast to a conventional HMM, the observations $(o_0, o_1, \ldots, o_V)$ are only available at irregularly-distributed continuous points in time $(t_0, t_1, \ldots, t_V)$. As a consequence, there are two levels of hidden information in a CT-HMM. First, at observation time, the state of the Markov chain is hidden and can only be inferred from measurements. Second, the state transitions in the Markov chain between two consecutive observations are also hidden. As a result, a Markov chain may visit multiple hidden states before reaching a state that emits a noisy observation. This additional complexity makes CT-HMM a more effective model for event data, in comparison to HMM and CTMC. But as a consequence the parameter learning problem is more challenging. We believe we are the first to present a comprehensive and systematic treatment of efficient EM algorithms to address these challenges.

A *fully observed* CT-HMM contains four sequences of information: the underlying state transition time $(t'_0, t'_1, \ldots, t'_{V'})$, the corresponding state $Y' = \{y_0 = s(t'_0), ..., y_{V'} = s(t'_{V'})\}$ of the hidden Markov chain, and the observed data $O = (o_0, o_1, \ldots, o_V)$ at time $T = (t_0, t_1, \ldots, t_V)$. Their joint complete likelihood can be written as

$$CL = \prod_{v'=0}^{V'-1} q_{y_{v'}, y_{v'+1}} e^{-q_{y_{v'}} \tau_{v'}} \prod_{v=0}^{V} p(o_v|s(t_v)) = \prod_{i=1}^{|S|} \prod_{j=1, j \neq i}^{|S|} q_{ij}^{n_{ij}} e^{-q_i \tau_i} \prod_{v=0}^{V} p(o_v|s(t_v)). \tag{4}$$

We will focus our development on the estimation of the transition rate matrix $Q$. Estimates for the parameters of the emission model $p(o|s)$ and the initial state distribution $\pi$ can be obtained from the standard discrete time HMM formulation [13], but with time-inhomogeneous transition probabilities (described below).

## 3.2 Parameter Estimation

Given a current estimate of the parameter $\hat{Q}_0$, the expected complete log-likelihood takes the form

$$L(Q) = \sum_{i=1}^{|S|} \sum_{j=1, j \neq i}^{|S|} \{\log(q_{ij})\mathbb{E}[n_{ij}|O, T, \hat{Q}_0] - q_i \mathbb{E}[\tau_i|O, T, \hat{Q}_0]\} + \sum_{v=0}^{V} \mathbb{E}[\log p(o_v|s(t_v))|O, T, \hat{Q}_0]. \quad (5)$$

In the M-step, taking the derivative of $L$ with respect to $q_{ij}$, we have

$$\hat{q}_{ij} = \frac{\mathbb{E}[n_{ij}|O, T, \hat{Q}_0]}{\mathbb{E}[\tau_i|O, T, \hat{Q}_0]}, i \neq j \quad \text{and} \quad \hat{q}_{ii} = -\sum_{j \neq i} \hat{q}_{ij}. \quad (6)$$

The challenge lies in the E-step, where we compute the expectations of $n_{ij}$ and $\tau_i$ conditioned on the observation sequence. The statistic for $n_{ij}$ can be expressed in terms of the expectations between successive pairs of observations as follows:

$$\mathbb{E}[n_{ij}|O, T, \hat{Q}_0] = \sum_{s(t_1),...,s(t_V)} p(s(t_1), ..., s(t_V)|O, T, \hat{Q}_0)\mathbb{E}[n_{ij}|s(t_1), ..., s(t_V), \hat{Q}_0] \quad (7)$$

$$= \sum_{s(t_1),...,s(t_V)} p(s(t_1), ..., s(t_V)|O, T, \hat{Q}_0) \sum_{v=1}^{V-1} \mathbb{E}[n_{ij}|s(t_v), s(t_{v+1}), \hat{Q}_0] \quad (8)$$

$$= \sum_{v=1}^{V-1} \sum_{k,l=1}^{|S|} p(s(t_v) = k, s(t_{v+1}) = l|O, T, \hat{Q}_0)\mathbb{E}[n_{ij}|s(t_v) = k, s(t_{v+1}) = l, \hat{Q}_0]. \quad (9)$$

In a similar way, we can obtain an expression for the expectation of $\tau_i$:

$$\mathbb{E}[\tau_i|O, T, \hat{Q}_0] = \sum_{v=1}^{n-1} \sum_{k,l=1}^{|S|} p(s(t_v) = k, s(t_{v+1}) = l|O, T, \hat{Q}_0)\mathbb{E}[\tau_i|s(t_v) = k, s(t_{v+1}) = l, \hat{Q}_0]. \quad (10)$$

In Section 4, we present our approach to computing the end-state conditioned statistics $\mathbb{E}[n_{ij}|s(t_v) = k, s(t_{v+1}) = l, \hat{Q}_0]$ and $\mathbb{E}[\tau_i|s(t_v) = k, s(t_{v+1}) = l, \hat{Q}_0]$. The remaining step is to compute the *posterior state distribution* at two consecutive observation times: $p(s(t_v) = k, s(t_{v+1}) = l|O, T, \hat{Q}_0)$.

## 3.3 Computing the Posterior State Probabilities

The challenge in efficiently computing $p(s(t_v) = k, s(t_{v+1}) = l|O, T, \hat{Q}_0)$ is to avoid the explicit enumeration of all possible state transition sequences and the variable time intervals between intermediate state transitions (from $k$ to $l$). The key is to note that the posterior state probabilities are only needed at the times where we have observation data. We can exploit this insight to reformulate the estimation problem in terms of an equivalent discrete time-*inhomogeneous* hidden Markov model.

Specifically, given the current estimate $\hat{Q}_0$, $O$ and $T$, we will divide the time into $V$ intervals, each with duration $\tau_v = t_v - t_{v-1}$. We then make use of the transition property of CTMC, and associate each interval $v$ with a state transition matrix $P^v(\tau_v) := e^{\hat{Q}_0 \tau_v}$. Together with the emission model $p(o|s)$, we then have a discrete time-inhomogeneous hidden Markov model with joint likelihood:

$$\prod_{v=1}^{V} [P^v(\tau_v)]_{(s(t_{v-1}), s(t_v))} \prod_{v=0}^{V} p(o_v|s(t_v)). \quad (11)$$

The formulation in Eq. 11 allows us to reduce the computation of $p(s(t_v) = k, s(t_{v+1}) = l|O, T, \hat{Q}_0)$ to familiar operations. The forward-backward algorithm [13] can be used to compute the posterior distribution of the hidden states, which we refer to as the *Soft* method. Alternatively, the MAP assignment of hidden states obtained from the Viterbi algorithm can provide an approximate distribution, which we refer to as the *Hard* method.

## 4 EM Algorithms for CT-HMM

Pseudocode for the EM algorithm for CT-HMM parameter learning is shown in Algorithm 1. Multiple variants of the basic algorithm are possible, depending on the choice of method for computing the end-state conditioned expectations along with the choice of *Hard* or *Soft* decoding for obtaining the posterior state probabilities in Eq. 11. Note that in line 7 of Algorithm 1,

---
**Algorithm 1** CT-HMM Parameter learning (Soft/Hard)
---
1: **Input:** data $O = (o_0, ..., o_V)$ and $T = (t_0, ..., t_V)$, state set $S$, edge set $L$, initial guess of $Q$
2: **Output:** transition rate matrix $Q = (q_{ij})$
3: Find all distinct time intervals $t_\Delta$, $\Delta = 1, ..., r$, from $T$
4: Compute $P(t_\Delta) = e^{Qt_\Delta}$ for each $t_\Delta$
5: **repeat**
6:      Compute $p(v, k, l) = p(s(t_v) = k, s(t_{v+1}) = l | O, T, Q)$ for all $v$, and the complete/state-optimized data likelihood $l$ by using Forward-Backward (soft) or Viterbi (hard)
7:      Create soft count table $C(\Delta, k, l)$ from $p(v, k, l)$ by summing prob. from visits of same $t_\Delta$
8:      Use *Expm, Unif* or *Eigen* method to compute $\mathbb{E}[n_{ij}|O, T, Q]$ and $\mathbb{E}[\tau_i|O, T, Q]$
9:      Update $q_{ij} = \frac{\mathbb{E}[n_{ij}|O,T,Q]}{\mathbb{E}[\tau_i|O,T,Q]}$, and $q_{ii} = -\sum_{i \neq j} q_{ij}$
10: **until** likelihood $l$ converges
---

we group probabilities from successive visits of same time interval and the same specified end-states in order to save computation time. This is valid because in a time-homogeneous CT-HMM, $\mathbb{E}[n_{ij}|s(t_v) = k, s(t_{v+1}) = l, \hat{Q}_0] = \mathbb{E}[n_{ij}|s(0) = k, s(t_\Delta) = l, \hat{Q}_0]$, where $t_\Delta = t_{v+1} - t_v$, so that the expectations only need to be evaluated for each distinct time interval, rather than each different visiting time (also see the discussion below Eq. 2).

### 4.1 Computing the End-State Conditioned Expectations

The remaining step in finalizing the EM algorithm is to discuss the computation of the end-state conditioned expectations for $n_{ij}$ and $\tau_i$ from Eqs. 9 and 10, respectively. The first step is to express the expectations in integral form, following [14]:

$$\mathbb{E}[n_{ij}|s(0) = k, s(t) = l, Q] = \frac{q_{i,j}}{P_{k,l}(t)} \int_0^t P_{k,i}(x) P_{j,l}(t-x)\, dx \tag{12}$$

$$\mathbb{E}[\tau_i|s(0) = k, s(t) = l, Q] = \frac{1}{P_{k,l}(t)} \int_0^t P_{k,i}(x) P_{i,l}(t-x)\, dx. \tag{13}$$

From Eq. 12, we define $\tau_{k,l}^{i,j}(t) = \int_0^t P_{k,i}(x) P_{j,l}(t-x) dx = \int_0^t (e^{Qx})_{k,i} (e^{Q(t-x)})_{j,l}\, dx$, while $\tau_{k,l}^{i,i}(t)$ can be similarly defined for Eq. 13 (see [6] for a similar construction). Several methods for computing $\tau_{k,l}^{i,j}(t)$ and $\tau_{k,l}^{i,i}(t)$ have been proposed in the CTMC literature. Metzner et. al. observe that closed-form expressions can be obtained when $Q$ is diagonalizable [8]. Unfortunately, this property is not guaranteed to exist, and in practice we find that the intermediate $Q$ matrices are frequently not diagonalizable during EM iterations. We refer to this approach as *Eigen*.

An alternative is to leverage a classic method of Van Loan [15] for computing integrals of matrix exponentials. In this approach, an auxiliary matrix $A$ is constructed as $A = \begin{bmatrix} Q & B \\ 0 & Q \end{bmatrix}$, where $B$ is a matrix with identical dimensions to $Q$. It is shown in [15] that $\int_0^t e^{Qx} B e^{Q(t-x)} dt = (e^{At})_{(1:n),(n+1):(2n)}$, where $n$ is the dimension of $Q$. Following [9], we set $B = I(i, j)$, where $I(i, j)$ is the matrix with a 1 in the $(i, j)$th entry and 0 elsewhere. Thus the left hand side reduces to $\tau_{k,l}^{i,j}(t)$ for all $k, l$ in the corresponding matrix entries. Thus we can leverage the substantial literature on numerical computation of the matrix exponential. We refer to this approach as *Expm*, after the popular Matlab function. A third approach for computing the expectations, introduced by Hobolth and Jensen [9] for CTMCs, is called *uniformization (Unif)* and is described in the supplementary material, along with additional details for *Expm*.

**Expm Based Algorithm** Algorithm 2 presents pseudocode for the *Expm* method for computing end-state conditioned statistics. The algorithm exploits the fact that the $A$ matrix does not change with time $t_\Delta$. Therefore, when using the *scaling and squaring* method [16] for computing matrix exponentials, one can easily cache and reuse the intermediate powers of $A$ to efficiently compute $e^{tA}$ for different values of $t$.

### 4.2 Analysis of Time Complexity and Run-Time Comparisons

We conducted asymptotic complexity analysis for all six combinations of *Hard* and *Soft* EM with the methods *Expm*, *Unif*, and *Eigen* for computing the conditional expectations. For both hard and

---

**Algorithm 2** The Expm Algorithm for Computing End-State Conditioned Statistics

---

1: **for** each state $i$ in $S$ **do**
2:     **for** $\Delta = 1$ **to** $r$ **do**
3:         $D_i = \frac{(e^{t_\Delta A})_{(1:n),(n+1):(2n)}}{P_{kl}(t_\Delta)}$, where $A = \begin{bmatrix} Q & I(i,i) \\ 0 & Q \end{bmatrix}$
4:         $\mathbb{E}[\tau_i | O, T, Q] += \sum_{(k,l) \in L} C(\Delta, k, l)(D_i)_{k,l}$
5:     **end for**
6: **end for**
7: **for** each edge $(i,j)$ in $L$ **do**
8:     **for** $\Delta = 1$ **to** $r$ **do**
9:         $N_{ij} = \frac{q_{ij}(e^{t_\Delta A})_{(1:n),(n+1):(2n)}}{P_{kl}(t_\Delta)}$, where $A = \begin{bmatrix} Q & I(i,j) \\ 0 & Q \end{bmatrix}$
10:         $\mathbb{E}[n_{ij} | O, T, Q] += \sum_{(k,l) \in L} C(\Delta, k, l)(N_{ij})_{k,l}$
11:     **end for**
12: **end for**

---

soft variants, the time complexity of *Expm* is $O(rS^4 + rLS^3)$, where $r$ is the number of distinct time intervals between observations, $S$ is the number of states, and $L$ is the number of edges. The soft version of *Eigen* has the same time complexity, but since the eigendecomposition of non-symmetric matrices can be ill-conditioned in any EM iteration [17], this method is not attractive. *Unif* is based on truncating an infinite sum and the truncation point $M$ varies with $\max_{i,t_\Delta} q_i t_\Delta$, with the result that the cost of *Unif* varies significantly with both the data and the parameters. In comparison, *Expm* is much less sensitive to these values (log versus quadratic dependency). See the supplemental material for the details. *We conclude that* Expm *is the most robust method available for the soft EM case*. When the state space is large, hard EM can be used to tradeoff accuracy with time. In the hard EM case, *Unif* can be more efficient than *Expm*, because *Unif* can evaluate only the expectations specified by the required end-states from the best decoded paths, whereas *Expm* must always produce results from all end-states.

These asymptotic results are consistent with our experimental findings. On the glaucoma dataset from Section 5.2, using a model with 105 states, *Soft Expm* requires 18 minutes per iteration on a 2.67 GHz machine with unoptimized MATLAB code, while *Soft Unif* spends more than 105 minutes per iteration, *Hard Unif* spends 2 minutes per iteration, and *Eigen* fails.

## 5 Experimental results

We evaluated our EM algorithms in simulation (Sec. 5.1) and on two real-world datasets: a glaucoma dataset (Sec. 5.2) in which we compare our prediction performance to a state-of-the-art method, and a dataset for Alzheimer's disease (AD, Sec. 5.3) where we compare visualized progression trends to recent findings in the literature. Our disease progression models employ 105 (Glaucoma) and 277 (AD) states, representing a significant advance in the ability to work with large models (previous CT-HMM works [2, 7, 5] employed fewer than 100 states).

### 5.1 Simulation on a 5-state Complete Digraph

We test the accuracy of all methods on a 5-state complete digraph with synthetic data generated under different noise levels. Each $q_i$ is randomly drawn from $[1, 5]$ and then $q_{ij}$ is drawn from $[0, 1]$ and renormalized such that $\sum_{j \neq i} q_{ij} = q_i$. The state chains are generated from $Q$, such that each chain has a total duration around $T = \frac{100}{\min_i q_i}$, where $\frac{1}{\min_i q_i}$ is the largest mean holding time. The data emission model for state $i$ is set as $N(i, \sigma^2)$, where $\sigma$ varies under different noise level settings. The observations are then sampled from the state chains with rate $\frac{0.5}{\max_i q_i}$, where $\frac{1}{\max_i q_i}$ is the smallest mean holding time, which should be dense enough to make the chain identifiable. A total of $10^5$ observations are sampled. The average 2-norm relative error $\frac{||\hat{q} - q||}{||q||}$ is used as the performance metric, where $\hat{q}$ is a vector contains all learned $q_{ij}$ parameters, and $q$ is the ground truth.

The simulation results from 5 random runs are listed in Table 1. *Expm* and *Unif* produce nearly identical results so they are combined in the table. *Eigen* fails at least once for each setting, but when it works it produces similar results. All *Soft* methods achieve significantly better accuracy

Table 1: The average 2-norm relative error from 5 random runs on a 5-state complete digraph under varying noise levels. The convergence threshold is $\leq 10^{-8}$ on relative data likelihood change.

| Error | $\sigma = 1/4$ | $\sigma = 3/8$ | $\sigma = 1/2$ | $\sigma = 1$ | $\sigma = 2$ |
|---|---|---|---|---|---|
| S(Expm,Unif) | 0.026±0.008 | 0.032±0.008 | 0.042±0.012 | 0.199±0.084 | 0.510±0.104 |
| H(Expm,Unif) | 0.031±0.009 | 0.197±0.062 | 0.476±0.100 | 0.857±0.080 | 0.925±0.030 |

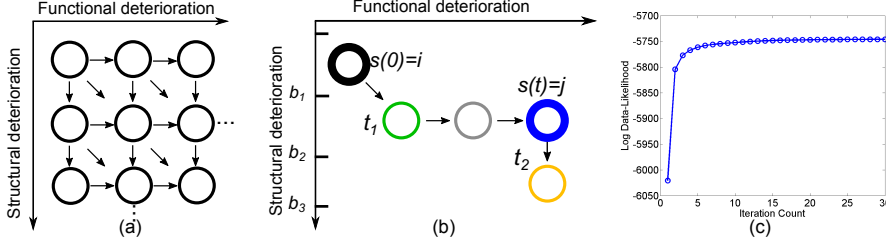

Figure 1: (a) The 2D-grid state structure for glaucoma progression modeling. (b) Illustration of the prediction of future states from $s(0) = i$. (c) One fold of convergence behavior of *Soft(Expm)* on the glaucoma dataset.

than *Hard* methods, especially when the noise level becomes higher. This can be attributed to the maintenance of the full hidden state distribution which makes it more robust to noise.

## 5.2 Application of CT-HMM to Predicting Glaucoma Progression

In this experiment we used CT-HMM to visualize a real-world glaucoma dataset and predict glaucoma progression. Glaucoma is a leading cause of blindness and visual morbidity worldwide [18]. This disease is characterized by a slowly progressing optic neuropathy with associated irreversible structural and functional damage. There are conflicting findings in the temporal ordering of detectable structural and functional changes, which confound glaucoma clinical assessment and treatment plans [19]. Here, we use a 2D-grid state space model with 105 states, defined by successive value bands of the two main glaucoma markers, Visual Field Index (VFI) (functional marker) and average RNFL (Retinal Nerve Fiber Layer) thickness (structural marker) with forwarding edges (see Fig. 1(a)). More details of the dataset and model can be found in the supplementary material. We utilize *Soft Expm* for the following experiments, since it converges quickly (see Fig. 1(c)), has an acceptable computational cost, and exhibits the best performance.

To predict future continuous measurements, we follow a simple procedure illustrated in Fig. 1(b). Given a testing patient, Viterbi decoding is used to decode the best hidden state path for the past visits. Then, given a future time $t$, the most probable future state is predicted by $j = \max_j P_{ij}(t)$ (blue node), where $i$ is the current state (black node). To predict the continuous measurements, we search for the future time $t_1$ and $t_2$ in a desired resolution when the patient enters and leaves a state having same value range as state $j$ for each disease marker separately. The measurement at time $t$ can then be computed by linear interpolation between $t_1$ and $t_2$ and the two data bounds of state $j$ for the specified marker ([$b1, b2$] in Fig. 1(b)). The mean absolute error (MAE) between the predicted values and the actual measurements was used for performance assessment. The performance of CT-HMM was compared to both conventional linear regression and Bayesian joint linear regression [11]. For the Bayesian method, the joint prior distribution of the four parameters (two intercepts and two slopes) computed from the training set [11] is used alongside the data likelihood. The results in Table 2 demonstrate the significantly improved performance of CT-HMM.

In Fig. 2(a), we visualize the model trained using the entire dataset. Several dominant paths can be identified: there is an early stage containing RNFL thinning with intact vision (blue vertical path in the first column), and at around RNFL range [80, 85] the transition trend reverses and VFI changes become more evident (blue horizontal paths). This $L$ shape in the disease progression supports the finding in [20] that RNFL thickness of around 77 microns is a tipping point at which functional deterioration becomes clinically observable with structural deterioration. Our 2D CT-HMM model can be used to visualize the non-linear relationship between structural and functional degeneration, yielding insights into the progression process.

## 5.3 Application of CT-HMM to Exploratory Analysis of Alzheimer's Disease

We now demonstrate the use of CT-HMM as an exploratory tool to visualize the temporal interaction of disease markers of Alzheimer's Disease (AD). AD is an irreversible neuro-degenerative disease that results in a loss of mental function due to the degeneration of brain tissues. An estimated 5.3

Table 2: The mean absolute error (MAE) of predicting the two glaucoma measures. (∗ represents that CT-HMM performs significantly better than the competing method under *student t-test*).

| MAE | CT-HMM | Bayesian Joint Linear Regression | Linear Regression |
|---|---|---|---|
| VFI | $4.64 \pm 10.06$ | $5.57 \pm 11.11$ * ($p = 0.005$) | $7.00 \pm 12.22$ *($p \approx 0.000$) |
| RNFL | $7.05 \pm 6.57$ | $9.65 \pm 8.42$ * ($p \approx 0.000$) | $18.13 \pm 20.70$ * ($p \approx 0.000$) |

million Americans have AD, yet no prevention or cures have been found [21]. It could be beneficial to visualize the relationship between clinical, imaging, and biochemical markers as the pathology evolves, in order to better understand AD progression and develop treatments.

A 277 state CT-HMM model was constructed from a cohort of AD patients (see the supplementary material for additional details). The 3D visualization result is shown in Fig. 2(b). The state transition trends show that the abnormality of $A\beta$ level emerges first (blue lines) when cognition scores are still normal. Hippocampus atrophy happens more often (green lines) when $A\beta$ levels are already low and cognition has started to show abnormality. Most cognition degeneration happens (red lines) when both $A\beta$ levels and Hippocampus volume are already in abnormal stages. Our quantitative visualization results supports recent findings that the decreasing of $A\beta$ level in CSF is an early marker before detectable hippocampus atrophy in cognition-normal elderly [22]. The CT-HMM disease model with interactive visualization can be utilized as an exploratory tool to gain insights of the disease progression and generate hypotheses to be further investigated by medical researchers.

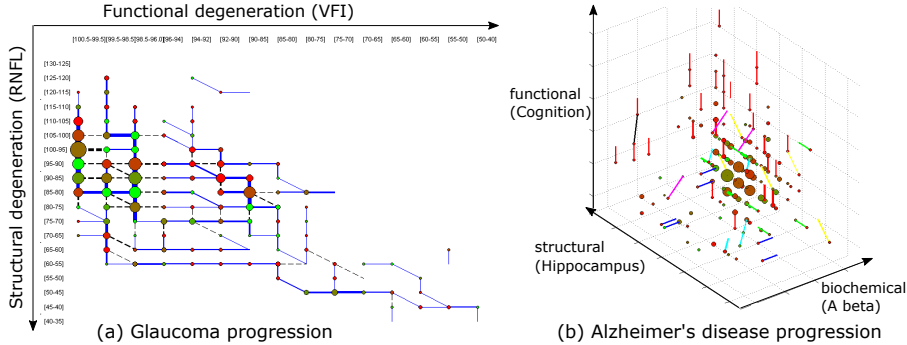

Figure 2: Visualization scheme: (a) The strongest transition among the three instantaneous links from each state are shown in blue while other transitions are drawn in dotted black. The line width and the node size reflect the expected count. The node color represents the average sojourn time (red to green: 0 to 5 years and above). (b) similar to (a) but the strongest transition from each state is color coded as follows: $A\beta$ direction (blue), $hippo$ (green), $cog$ (red), $A\beta + hippo$ (cyan), $A\beta + cog$ (magenta), $hippo + cog$ (yellow), $A\beta + hippo + cog(black)$. The node color represents the average sojourn time (red to green: 0 to 3 years and above).

## 6 Conclusion

In this paper, we present novel EM algorithms for CT-HMM learning which leverage recent approaches [9] for evaluating the end-state conditioned expectations in CTMC models. To our knowledge, we are the first to develop and test the *Expm* and *Unif* methods for CT-HMM learning. We also analyze their time complexity and provide experimental comparisons among the methods under soft and hard EM frameworks. We find that soft EM is more accurate than hard EM, and *Expm* works the best under soft EM. We evaluated our EM algorithsm on two disease progression datasets for glaucoma and AD. We show that CT-HMM outperforms the state-of-the-art Bayesian joint linear regression method [11] for glaucoma progression prediction. This demonstrates the practical value of CT-HMM for longitudinal disease modeling and prediction.

**Acknowledgments**

Portions of this work were supported in part by NIH R01 EY13178-15 and by grant U54EB020404 awarded by the National Institute of Biomedical Imaging and Bioengineering through funds provided by the Big Data to Knowledge (BD2K) initiative (www.bd2k.nih.gov). Additionally, the collection and sharing of the Alzheimers data was funded by ADNI under NIH U01 AG024904 and DOD award W81XWH-12-2-0012. The research was also supported in part by NSF/NIH BIGDATA 1R01GM108341, ONR N00014-15-1-2340, NSF IIS-1218749, and NSF CAREER IIS-1350983.

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
