[Supplementary Material]

Supplementary Material for
"Efficient Learning of Continuous-Time Hidden
Markov Models for Disease Progression",
in NIPS 2015

# 1   Additional Motivation for Using Continuous-time Models

A question may arise as to why continuous time models are needed for disease progression modeling. For example, why not simply discretize the time interval and apply discrete time models? The answer is that the time horizon for state changes in medical conditions can vary dramatically. In the early stages of a disease, a state change might not occur for years, but in an acute phase they could occur very frequently. For states with very short expected dwelling times, the time step in a discrete model would needs to be sufficiently small. However, this might be very inefficient for dealing with changes that occur over long time intervals. On the other hand, if the discretization is too coarse, many transitions could be collapsed into a single one, obscuring the real dynamics in continuous-time. In contrast, CT model performs inference over arbitrary timescales using a single matrix exponential. We believe CT is a better choice than DT for modeling continuous-time processes such as disease progression and clinical data.

# 2   Relationship to Sampling-Based Methods

Here we briefly discuss the relationship between our approach and that of [1]. In [1], CTMC trajectories are sampled from the posterior $p(S|O)$ ($S$: a CTMC trajectory, $O$: observations) using MCMC sampling based on *uniformization*[2] ideas. Its time complexity is dependent on $max_i q_i$, the fastest transition rate in the system, which suffers from the same discretization issue discussed earlier (which is also noted by the authors). The benefit of [1] over our *Expm* approach is that the time complexity only has quadratic dependency on the state space size, rather than cubic when matrix exponential operation is ever used. However, this benefit could be offset by the number of samples required for an accurate estimate, and the aforementioned discretization issue. While [1] does not address

parameter learning, their method could in principle be used as a step in our EM algorithm for computing the required statistics. However, our *Expm* approach computes those statistics directly and is more robust to varying transition rates. The exact comparison of which method is better is dependent on the distribution on the transition time, as well as the desired accuracy.

# 3 Additional Discussion of Methods for Computing End-State Conditioned Statistics

## 3.1 The Uniformization Method

The uniformization method (*Unif*) is an efficient approximation method for computing matrix exponential $P(t) = e^{Qt}$ [3, 4], which gives an alternative description of the CTMC process, and show how CTMC and DTMC is equivalence subordinated to a *Poisson* process (see [2]). Define $\hat{q} = \max_i q_i$, and matrix $R = \frac{Q}{\hat{q}} + I$, where $I$ is the identify matrix. Then, $e^{Qt} = e^{\hat{q}(R-I)t} = \sum_{m=0}^{\infty} R^m \frac{(\hat{q}t)^m}{m!} e^{-\hat{q}t} = \sum_{m=0}^{\infty} R^m Pois(m; \hat{q}t)$, where $Pois(m; \hat{q}t)$ is the probability of $m$ occurrences from a Poisson distribution with mean $\hat{q}t$. Then the expectations can be expressed by directly inserting the $e^{Qt}$ series into the integral: $E[\tau_i, s(t) = l|s(0) = k] = \sum_{m=0}^{\infty} \frac{t}{m+1}[\sum_{n=0}^{m}(R^n)_{ki}(R^{m-n})_{il}]Pois(m; \hat{q}t)$ and $E[n_{ij}, s(t) = l|s(0) = k] = R_{ij}\sum_{m=1}^{\infty}[\sum_{n=1}^{m}(R^{n-1})_{ki}(R^{m-n})_{jl}]Pois(m; \hat{q}t)$ [4]. The main difficulty in using *Unif* in practice is to determine a truncation point of the infinite sum. However, for large values of $\hat{q}t$, we have $Pois(\hat{q}t) \approx N(\hat{q}t, \hat{q}t)$, where $N(\mu, \sigma^2)$ is the normal distribution and one can then bound the truncation error from the tail of Poisson by using cumulative normal distribution [5]. A truncation point at $M = \lceil 4 + 6\sqrt{\hat{q}t} + (\hat{q}t) \rceil$ is suggested [5] to have error bound of $10^{-8}$ when approximate $P(t)$, which we adopt in our learning algorithm 1.

## 3.2 The Eigendecomposition Method

To compute $\tau_{k,l}^{i,i}(t)$ and $\tau_{k,l}^{i,j}(t)$, it is observed in [6] that the calculation of $\tau_{k,l}^{i,j}(t)$ can be done in closed-form if $Q$ is diagonalizable and one can act eigendecomposition on $Q$ (*Eigen* method). Consider the eigendecomposition of $Q = UDU^{-1}$, where the matrix $U$ consists of all eigenvectors to the corresponding eigenvalues of $Q$ in the diagonal matrix $D = diag(\lambda_1, ..., \lambda_n)$. Then we have $e^{Qt} = Ue^{Dt}U^{-1}$ and the integral can be written as: $\tau_{k,l}^{i,j}(t) = \sum_{p=1}^{n} U_{kp}U_{pi}^{-1}\sum_{q=1}^{n} U_{jq}U_{ql}^{-1}\Psi_{pq}(t)$ ,where the symmetric matrix $\Psi(t) = [\Psi_{pq}(t)]_{p,q \in S}$ is defined as: $\Psi_{pq}(t) = te^{t\lambda_p}$ if $\lambda_p = \lambda_q$, and $\Psi_{pq}(t) = \frac{e^{t\lambda_p} - e^{t\lambda_q}}{\lambda_p - \lambda_q}$ if $\lambda_p \neq \lambda_q$.

# 4 Additional EM Algorithms for CT-HMM Learning

## 4.1 Unif Based Algorithm

We used *Unif* method for computing end-state conditioned statistic for CTMC in Algorithm 1. In line 6 and 10, $S_{k\to l}$ (and $L_{k\to l}$) represents the intermediate states (edges) that can be passed from state $k$ to $l$. The state accessibility table can be precomputed using *Dijkastra*'s shortest path algorithm in $O(S^2)$. The main benefits of *Unif* in evaluating all expectations is that the $R$ series $(R, R^2, ..., R^{\hat{M}})$, can be precomputed (line 2) and reused, so that no additional matrix multiplications is needed. One main property of *Unif* is that it can evaluate the expectations for only the two specified end-states, and it has $O(M^2)$ complexity, which is not related to $S$ (when given the precomputed $R$ matrix series). In hard EM the soft count table $C(\Delta, k, l)$ (in line 5) becomes sparse ($\leq min(V, rS^2)$ entries have positive values), and thus *Unif* in hard EM becomes more time efficient than soft EM. One possible downside of *Unif* is that if $\hat{q}_i t$ is very large, so is the truncation point $M$, then the computation can be very time consuming. Thus, we find that *Unif*'s running time performance highly depends on the data and the underlying Q values. The time complexity analysis is detailed in Algorithm 1 line 16.

---

**Algorithm 1** Unif Algorithm

---

1: Set $\hat{t} = max\ t_\Delta$; set $\hat{q} = max_i q_i$.
2: Let $R = Q/\hat{q} + I$. Compute $R, R^2, ..., R^{\hat{M}}$, $\hat{M} = \lceil 4 + 6\sqrt{\hat{q}\hat{t}} + (\hat{q}\hat{t}) \rceil \Rightarrow O(\hat{M}S^3)$

3: **for** $\Delta = 1$ **to** $r$ **do**
4:    $M = \lceil 4 + 6\sqrt{\hat{q}t_\Delta} + (\hat{q}t_\Delta) \rceil$; set $t = t_\Delta$
5:    **for** each $C(\Delta, k, l) \neq 0$ **do**
6:       **for** each state $i$ in $S_{k\to l}$ **do**
7:          $E[\tau_i|s(0) = k, s(t) = l, Q] = \frac{\sum_{m=0}^{M} \frac{t}{m+1}[\sum_{n=0}^{m}(R^n)_{ki}(R^{m-n})_{il}]Pois(m;\hat{q}t)}{P_{kl}(t)}$
         $\Rightarrow O(M^2)$
8:          $E[\tau_i|O, T, Q] += C(\Delta, k, l)E[\tau_i|s(0) = k, s(t) = l]$
9:       **end for**
10:      **for** each edge $(i, j)$ in $L_{k\to l}$ **do**
11:         $E[n_{ij}|s(0) = k, s(t) = l, Q] = \frac{R_{ij}\sum_{m=1}^{M}[\sum_{n=1}^{m}(R^{n-1})_{ki}(R^{m-n})_{jl}]Pois(m;\hat{q}t)}{P_{kl}(t)} \Rightarrow O(M^2)$
12:         $E[n_{ij}|O, T, Q] += C(\Delta, k, l)E[n_{ij}|s(0) = k, s(t) = l]$
13:      **end for**
14:    **end for**
15: **end for**
16: Soft:   $O(\hat{M}S^3 + rS^3M^2 + rS^2LM^2)$;   Hard:   $O(\hat{M}S^3 + min(V, rS^2)SM^2 + min(V, rS^2)LM^2)$

---

## 4.2 Eigendecomposition-Based Algorithm

The algorithm with time complexity is listed in Algorithm 2 and Algorithm 3, where the latter one can be more efficient for soft-EM learning. *Eigen* method also has the flexibility in evaluating the expectations only for the specified end-states, and thus it can be more efficient in hard than in soft EM. In soft EM learning where we need to compute expectations for every possible end-state pairs, one can compute the unknowns simultaneously via matrix multiplications [7] to be more cost efficient than just compute for each unknown separately. In more detail, define the matrix $D_i$, where the $(k, l)$ entry is $E[\tau_i|s(0) = k, s(t) = l]$. Let $U_i^{-1}$ represent the $i$th row of the matrix $U^{-1}$, and $U_i$ represent the $i$'s column of $U$. Then [7] shows that $D_i = U[(U_i^{-1}U_i) * \Psi]U^{-1}$, where $A * B$ is the entrywise product of the two matrices $A$ and $B$. Using this relationship, a total of $O(S^3)$ is achieved to compute $E[\tau_i|s(0) = k, s(t) = l]$ for all $k, l$ for a fixed $i$. Similarly, we can define the matrix $N_{ij}$, where the $(k, l)$ entry is $E[n_{ij}|s(0) = k, s(t) = l]$ and uses the equation $N_{ij} = q_{ij}U[(U_i^{-1}U_j) * \Psi]U^{-1}$, to compute the unknowns efficiently.

The main problem of *Eigen* is that it is not a stand-alone algorithm. When $Q$ is not diagonalizable in any iteration, one needs alternative methods for that run. In addition, the eigendecomposition of non-symmetric matrices can be ill-conditioned [7], and one needs reliable numerical solver to indicate this and uses other approaches.

---

**Algorithm 2** Eigen Algorithm (Soft/Hard EM)

---

1: Perform eigendecomposition: $Q = UDU^{-1} \Rightarrow O(S^3)$
2: **for** $\Delta = 1$ **to** $r$ **do**
3:    Compute matrix $\Psi$ with $t = t_\Delta \Rightarrow O(S^2)$
4:    **for** each $C(\Delta, k, l) \neq 0$ **do**
5:       **for** each state $i$ in $S_{k \to l}$ **do**
6:          $E[\tau_i|s(0) = k, s(t) = l, Q] = \frac{\sum_{p=1}^{|S|} U_{kp}U_{pi}^{-1} \sum_{q=1}^{|S|} U_{iq}U_{ql}^{-1}\Psi_{pq}(t)}{P_{kl}(t)} \Rightarrow O(S^2)$
7:          $E[\tau_i|O, T, Q] + = C(\Delta, k, l)E[\tau_i|s(0) = k, s(t) = l]$
8:       **end for**
9:       **for** each link $(i, j)$ in $L_{k \to l}$ **do**
10:         $E[n_{ij}|s(0) = k, s(t) = l, Q] = q_{ij}\frac{\sum_{p=1}^{|S|} U_{kp}U_{pi}^{-1} \sum_{q=1}^{|S|} U_{jq}U_{ql}^{-1}\Psi_{pq}(t)}{P_{kl}(t)} \Rightarrow O(S^2)$
11:         $E[n_{ij}|O, T, Q] + = C(\Delta, k, l)E[n_{ij}|s(0) = k, s(t) = l]$
12:      **end for**
13:   **end for**
14: **end for**
15: Soft: $O(rS^5 + rLS^4)$; Hard: $O(min(rS^2, V)S^3 + min(rS^2, V)LS^2)$

---

**Algorithm 3** Eigen Algorithm (Soft-EM)

---

1: Perform eigendecomposition: $Q = UDU^{-1} \Rightarrow O(S^3)$
2: **for** $\Delta = 1$ **to** $r$ **do**
3:     Compute matrix $\Psi$ with $t = t_\Delta \Rightarrow O(S^2)$
4:     **for** each state $i$ in $S$ **do**
5:         $D_i = U[(U_i^{-1}U_i) * \Psi]U^{-1} \Rightarrow O(S^3)$
6:         $E[\tau_i|O,T,Q] + = \sum_{(k,l)\in L} C(\Delta,k,l)(D_i)_{k,l}$
7:     **end for**
8:     **for** each edge $(i,j)$ in $L$ **do**
9:         $N_{ij} = q_{ij}U[(U_i^{-1}U_j) * \Psi]U^{-1} \Rightarrow O(S^3)$
10:         $E[n_{ij}|O,T,Q] + = \sum_{(k,l)\in L} C(\Delta,k,l)(N_{ij})_{k,l}$
11:     **end for**
12: **end for**
13: Soft: $O(rS^4 + rLS^3)$

---

# 5 Time Complexity Comparison between Expm and Other Methods

Table 1: Time complexity comparison of all methods in evaluating all required expectations under Soft/Hard EM ($r$: number of distinct time interval, $S$: number of states, $L$: number of edges, $V$: number of visits, $M$: the largest truncation point of the infinite sum for *Unif*, set as $\lceil 4 + 6\sqrt{\hat{q}\hat{t}} + (\hat{q}\hat{t}) \rceil$, where $\hat{q} = \max_i q_i$, and $\hat{t} = max_\Delta t_\Delta$).

| complexity | Expm | Unif | Eigen |
|---|---|---|---|
| Soft EM | $O(rS^4 + rLS^3)$ | $O(MS^3 + rS^3M^2 + rS^2LM^2)$ | $O(rS^4 + rLS^3)$ |
| Hard EM | $O(rS^4 + rLS^3)$ | $O(MS^3 + \min(rS^2,V)SM^2$ $+ \min(rS^2,V)LM^2)$ | $O(\min(rS^2,V)S^3$ $+ \min(rS^2,V)LS^2$ |

See Table 1 for a comparison of time complexities among the three methods. The time complexity of *Expm* is less sensitive to $\max_i q_i t_\Delta$ than *Unif* method (*log* versus *quadratic* dependency). It is because when *Expm* is evaluated using the *scaling and squaring* method [8], the number of matrix multiplications depends on the number of doing matrix scaling and squaring, which is $\lceil \log_2(||Qt_\Delta||_1/\theta_{13}) \rceil$, where $\theta_{13} = 5.4$ (the *Pade* approximant with degree 13), if scaling of $Q$ is required [8]. Then we have $\log_2(||Qt_\Delta||_1) \leq \log_2(S \max_i q_i t_\Delta)$. Thus, the running time of *Unif* will change according to $\max q_i t$ more dramatically than *Expm* method.

When comparing Soft EM methods, we find that *S(Expm)* and *S(Eigen)* have same order of time complexity. However, *Eigen* is not a stand-alone algorithm. When $Q$ is not diagonalizable or the eigendecomposition of non-symmetric matrices is ill-conditioned in any iteration [7], one needs alternative methods. The time complexity comparison between *Expm* and *Unif* depends on the relative scale between state space $S$ and $M^2$, where $M = \lceil 4 + 6\sqrt{\max_i q_i t_\Delta} + (\max_i q_i t_\Delta) \rceil$. The time complexity of *Expm* is less sensitive to $\max_i q_i t_\Delta$ than

*Unif* method (*log* versus *quadratic* dependency).

# 6 Details in the Glaucoma and Alzheimer's Experiments

## 6.1 The Glaucoma Experiment

Our glaucoma dataset contains 101 glaucomatous eyes from 74 patients followed for an average of 11.7±4.5 years, and each eye has at least 5 visits (average 7.1±3.1 visits). 63 distinct time intervals are found. The state space is created so that most states have at least 5 raw measurements mapped to it. The states which are in the straight path in between two successive raw data are instantiated, resulting in 105 states [1] . The data emission model is set as a normal distribution with $\mu$ set to the center of the data band, and $\sigma$ set to 0.25 of the band width. Ten-fold cross validation is used and *Soft(Expm)* is adopted in learning. Testing proceeds by decoding the first 4 visits using the learned CT-HMM model and then predicting future states and observations.

## 6.2 The Alzheimers Dataset

In this experiment, we analyze the temporal interaction among the three kinds of markers: amyloid beta ($A\beta$) level in cerebral spinal fluid (CSF) (bio-chemical marker), hippocampus volume (structural marker), and ADAS cognition score (functional marker) over the course of the disease. We obtained the *ADNI* (The *Alzheimer's Disease Neuroimaging Initiative*) dataset from the website [9][2] The mild cognition impairment (MCI) and AD patients who have at least two visits of all three indicated markers are included for our analysis, which results in 206 subjects of $2.38 \pm 0.66$ visits traced in $1.56 \pm 0.86$ years. Only 3 distinct time intervals in month resolution are found. A 3D gridded state space with forwarding links is defined such that for each marker, we have 14 bands that span its value range. The procedure for constructing the state space and the definition of data emission model is the same as in the Glaucoma experiment. 277 states are instantiated and the model is then trained using *Soft(Expm)*. The running time using *Soft(Expm)* is about 17 minutes per iteration on a 2.67 GHz machine (for comparison, *Soft(Unif)* spends more than 48 minutes per iteration; *Hard(Unif)* spends around 2 minutes and *Eigen* fails in this model).

## Footnotes

[1] The grid [100.5 99.5 98 96 93 90 85 80 : (−10) : 20] is used for Visual Field Index (the functional marker), and the grid [130 : (−5) : 80 70 : (−10) : 30] is used for the Retinal Nerve Fiber Layer thickness (the structural marker) for the glaucoma prediction task.

[2]Data were obtained from the ADNI database (`adni.loni.usc.edu`). The ADNI was launched in 2003 as a public-private partnership, led by Principal Investigator Michael W. Weiner, MD. The primary goal of ADNI has been to test whether serial magnetic resonance imaging (MRI), positron emission tomography (PET), other biological markers, and clinical and neuropsychological assessment can be combined to measure the progression of mild cognitive impairment (MCI) and early Alzheimers disease (AD). For up-to-date information, see `http://www.adni-info.org`