[Reviews · NeurIPS 2015]

Submitted by Assigned_Reviewer_1

The paper presents several EM-style algorithms for learning Continuous-Time Hidden Markov Models (CT-HMMs) and illustrates them on two disease evolution prediction tasks (Glaucoma and Alzheimer's). The paper motivation is nicely laid out and this framework is indeed more realistic than the usual discrete-time HMM in practical applications such as medicine and customer interaction. The framework of CT-HMM is clearly explained and the derivations of the algorithms appear to be correct. From a theoretical point of view, the approach is straightforward but the technical details for getting the algorithms are not trivial, so this makes a nice theoretical contribution. In terms of the empirical evaluation, it is nice that the authors work with real data sets, and this strengthens the paper. However, on the Glaucoma dataset, the comparison methods are very weak (linear and Bayesian linear regression). The more interesting comparison would be against a discrete-time HMM (one which ignores the elapsed time between visits). This comparison should be added to the paper.

The Alzheimer;'s results are more qualitative but it is nice to have a second task. Sec. 5 needs to be revised from the point of view of the writing, as it has a lot more unclear sentences and grammar errors than the rest of the paper. As a side note, using only exponential distributions for the transitions times, while standard in CTMCs, is not always best in practical terms - mixtures may be needed. This might be interesting for future work.
Summary: The authors derive EM-style algorithms for learning Continuous-Time Hidden Markov Models (in which state transitions may occur between observations). The paper makes a nice theoretical contribution. The empirical evaluation could be improved, but the use of real data makes the work worthy of publication.

Submitted by Assigned_Reviewer_2

I wonder what trade-off the discretized time interval (Sec 3.1) introduces, is this exact? Analogously, the same discretization in time can be applied to CTMC, what will this do?

The writing of this paper is quite dense overall.

I feel that too much space is spent explaining prior art (until line 235, 4.5 pages into the paper).

In sec 5.1 results with simulated data - the mean of the observations from each state is 1.0 apart, with standard deviation \sigma=0.25. With states 4\sigma apart, the results from this section has hardly any noise. I wonder how this result helps show the efficacy of the proposed scheme, as opposed to CTMC.

Furthermore, the error metric "2-norm error" is not defined, assume this is on the state sequence? as said above, where could the errors in states have come from?

In section 2 background, it is better to clarify the contributions of [8,9,10,12,13] relative to each other, it will help the reader conceptually appreciate this topic more.

One recent paper should have been cited and commented on: V. Rao and Y. W. Teh. Fast MCMC sampling for Markov jump processes and extensions. Journal of Machine Learning Research, 13, 2014.
Summary: This paper proposes two inference schemes for continuos-time hidden markov models (CT-HMM) by extending recent inference methods for continuous-time markov chains (CTMC). The algorithmic contribution is a little bit novel. Applications relevant and somewhat novel (to the NIPS community, from disease progression modeling).

The main concerns are in writing quality and quality of results.

Submitted by Assigned_Reviewer_3

The paper presents new methods to learn parameters and then perform inference (either Viterbi or forward/backward) for continous time HMM models (CT-HMM) with non uniformly distributed time points. They solve the inference problem for the transition rate matrix Q by constructing a discrete time inhomogeneous HMM then applying two recently developed methods to get sufficient statistics for their EM (Expfm, Unif - from Hobolth & Jensen 2011). They follow with a detailed time complexity analysis for the different flavors of the their algorithm (Table 1) before proceeding to experimental evaluation. Experiments include a 5 state synthetic dataset and two disease progression datasets for Glaucoma and AD.

Pros: The paper is mostly well written and the problem addressed is interesting in terms of the computational challenge, the modeling aspects, and the biomedical domain. The authors provide mathematical foundation for their approach building on previous work by

Hobolth & Jensen 2011 for the specific task at hand. The authors do a good job explaining the biomedical datasets/questions and relating their results to what is known about the disease.

Cons: 1. The synthetic data evaluation is not sufficient. Why 5 states? 10^6 samples seems an awful lot of samples to train on an not representative of real life problems. The difference between states emission (line 319-320) seems too easy? That may explain why all their algorithm's variants seem to preform similarly. Harder settings and varying different parameters may flush out differences better and give a bette sense of how the algorithm can perform on real life data. 2. Sec. 5.2 is rather cryptic, especially the description of the competing methods. Consider adding details about those in the appendix to make this section more clear.

Minor comments: 1. Fig2 is not self contained. Referring to it first (line 40) is premature. 2. Line 91 "the rate the process" 3. Line 161 "in only at" 4. Sec 5.2: the claim about binary search (lines 348-350) is not clear. 5. Line 399 "to shown"
Summary: The paper develop new methods to efficiently perform parameter learning and subsequent inference for CT-HMM, then apply the method to two disease progression datasets.

Submitted by Assigned_Reviewer_4

This paper builds upon previous EM method for CTMC, and tackles the conditional statistics borrowing some recently developed tools. It makes a detailed comparison over a few inference methods, in terms of accuracy and complexity.

Quality: a. For experiments, the baselines chosen are too week since they do not consider state transition/trajectory information. It is natural to compare with discrete time HMM approaches where you can set discretize the time horizon by setting a proper bin size.

b. Related to a., considering the time complexity to do inference on CT-HMM, why not simply discretize the time horizon? If the sampling rate is irregular, we can consider a proper bin size: if bin size is small, we have 1 or 0 (missing) observation in each bin; if bin size is large, we have >=1 observations in each bin and take a local average. I expect in most problems, this method works just as fine. Have you considered this approach for your dataset? c. For CT-HMM, other inference methods like sampling also exist, i.e., Fast MCMC sampling for Markov jump processes and extensions, Rao et al., JMLR 2013. Have you also compared with such approaches?

Clarity: The paper is clearly written.

Originality: This is novel in the sense it provides an EM-based approach for CT-HMM, although the tools/building blocks used were proposed in previous work.

Significance: The paper proposes an EM-based inference method for CT-HMM. If the author can release their software package, it will be a useful toolbox for the community.
Summary: This work presents an EM method for CT-HMM model, using some of recently developed tools. Though incremental, it provides an inference method for CT-HMMs.

Author Feedback
Author rebuttal: We thank the reviewers for their positive remarks on our paper's novelty, practicality as a useful toolbox, good writing and having contributions on both methodological and application aspects.

We present the first comprehensive characterization of EM-based learning methods in CT-HMM and demonstrate their use to visualize and predict future disease measurements for Glaucoma and Alzheimer's disease. The novel EM algorithms scale significantly better than prior art, and can capture complex dynamics of practical longitudinal disease evolution. Our final version will include more synthetic experiments, offer more intuitions in the writing, and reflect the comments such as reducing the space for discussing prior art.

R1,R3: For synthetic data, harder settings and varying different parameters may flush out differences better.
A: New synthetic experiments are conducted with different noise levels, and the observation count is reduced from 10^6 to 10^5 to be more realistic. Five different standard deviations in the data emission model are tested with 5 random runs each:

sigma=1/4,sigma = 3/8,sigma=1/2,sigma=1,sigma=2

S(Expm), S(Unif): 0.0261+-0.0080, 0.0324+-0.0080, 0.0420+-0.0118, 0.1990+-0.0835, 0.5096+-0.1037

H(Expm),H(Unif): 0.0314+-0.0089, 0.1968+-0.0622, 0.4762+-0.0995, 0.8568+-0.0801, 0.9249+-0.0298

Eigen fails at least once for each setting (but when it works, it produces similar results as the other two). The results show that soft methods are more robust than hard ones.

R3: Why not simply discretize the time horizon?
A: The time horizon for state changes in medical conditions can vary dramatically. In early stages of disease a state change might not occur for years, but in an acute phase they could occur very frequently. For states with very short expected dwelling times, the discrete time step needs to be sufficiently small. However, this might be inefficient for dealing with changes that occur once several years. On the other hand, if the discretization is too coarse, many transitions could be collapsed into a single one, obscuring the real dynamics in continuous-time. In contrast, CT model performs inference over arbitrary timescales using a single matrix exponential. We believe CT is a better choice than DT for modeling continuous-time processes such as disease progression and clinical data.

R3,R5: Compare to (Rao et al., JMLR 2013)?
A: This relevant reference will be cited. [Rao] samples CTMC trajectories from the posterior p(S|O) (S: a CTMC trajectory, O: observations) using MCMC sampling based on uniformization. Its time complexity is dependent on max_i q_i, the fastest transition rate in the system, which suffers from the same discretization issue as in we mentioned in the previous answer (the authors of [Rao] also noted that in the paper). The benefit of [Rao] over our Expm approach is that the time complexity only has quadratic dependency on the state space size, rather than cubic as in Expm. However, this could be offset by the number of samples required for an accurate estimate, and the aforementioned discretization issue.

Besides, [Rao] didn't talk about parameter learning, which is an important part of our work. Presumably, their method could be used as a step in our EM algorithm for computing the required statistics. However, our Expm approach computes those statistics and is more robust to varying transition times. The exact comparison of which method is better is dependent on the distribution on the transition time, as well as the desired accuracy.

R5: What tradeoff the discretized time interval (Sec 3.1) introduces, is this exact?
A: It is exact and efficient. The time inhomogeneous transition matrix P(tau_v) = exp(Q tau_v) between two visits summarizes all possible hidden transition paths and times between every end-state pair given a time interval, and the dynamic programming efficiently summarizes probabilities of all possible hidden states at observation times. (For CTMC learning, this inference is not required as the states are directed observed.)

R5: The error metric "2norm error"?
A: This assesses the learning accuracy of qij parameters (e.g. there are 20 qij parameters for the 5-state complete digraph). Suppose v_1 is a vector of the estimated qij and v_2 represents the ground truth parameters. The relative 2-norm error = ||v_1 - v_2||/||v_2|| using a vector 2-norm.

R7: You might benefit from looking into the field of "continuous-time system identification":
A: We checked upon the request but haven't found prior work in this field that solves efficient parameter learning for large CT-HMM. Most continuous-time system identification papers work on input-output models, quite different from CT-HMM. Note our approach is not the first one on CT-HMM, however existing methods either can't scale up well or have restrictive assumptions on the timing of state transitions. Ours lift those restrictions.